# Splicing Variants, Protein-Protein Interactions, and Drug Targeting in Hutchinson-Gilford Progeria Syndrome and Small Cell Lung Cancer

**DOI:** 10.3390/genes13020165

**Published:** 2022-01-18

**Authors:** Bae-Hoon Kim, Tae-Gyun Woo, So-Mi Kang, Soyoung Park, Bum-Joon Park

**Affiliations:** 1Rare Disease R&D Center, PRG S&T Co., Ltd., Busan 46241, Korea; bk728@prgst.com (B.-H.K.); taegyun0728@prgst.com (T.-G.W.); 2Department of Molecular Biology, College of Natural Science, Pusan National University, Busan 46274, Korea; rosa.somi.kang@pusan.ac.kr (S.-M.K.); thdud2971@naver.com (S.P.)

**Keywords:** Hutchinson-Gilford progeria syndrome (HGPS), small cell lung cancer (SCLC), alternative splicing (AS), Progerin, DX2, protein-protein interactions (PPIs)

## Abstract

Alternative splicing (AS) is a biological operation that enables a messenger RNA to encode protein variants (isoforms) that give one gene several functions or properties. This process provides one of the major sources of use for understanding the proteomic diversity of multicellular organisms. In combination with post-translational modifications, it contributes to generating a variety of protein–protein interactions (PPIs) that are essential to cellular homeostasis or proteostasis. However, cells exposed to many kinds of stresses (aging, genetic changes, carcinogens, etc.) sometimes derive cancer or disease onset from aberrant PPIs caused by DNA mutations. In this review, we summarize how splicing variants may form a neomorphic protein complex and cause diseases such as Hutchinson-Gilford progeria syndrome (HGPS) and small cell lung cancer (SCLC), and we discuss how protein–protein interfaces obtained from the variants may represent efficient therapeutic target sites to treat HGPS and SCLC.

## 1. Introduction

Around 44 years ago, Joseph Sambrook discovered the exon and intron in a gene from an adenovirus, and Walter Gilbert suggested that different exon fusions in a gene might be spliced together to make different mRNA isoforms [1,2]. As a result of scientists’ efforts (the Human Genome Project) to create a global view of genomes, 20 years ago, scientists have calculated that an average human protein-coding gene has 8.8 exons, with an average of 145 nt. The intron length and the lengths of the 5′ and 3′ UTR are, on average 3365 nt, 770 nt, and 300 nt, respectively [3]. Most protein-coding genes have introns that are degraded in the nucleus by the RNA splicing machinery. Exon usage is very often alternative. The cell decides to get rid of introns from the pre-mRNA or include exon parts in the mature mRNA [4]. There are around 20,000 human gene-encoding proteins and around 150,000 isoforms from RNA transcripts. Consequently, in general, each human gene has about seven RNA isoforms. Recently, it was reported that over 30% of tissue-dependent transcriptional variations are covered by local splicing variations, which are described as a split in a splice graph into or from a single exon [5]. The resulting differences among the mature mRNA isoforms produced by alternative splicing (AS) mean they can encode different protein products, or affect the localization, stability, and translation of mRNA. When we consider that the high level of complexity in the human genome may be illustrated by the variability of each gene with AS, it is natural to think of the fine-tuning control of these as necessary and flexibility as risky. Considering the importance of RNA splicing for protein diversity and keeping the function of the organism, it is clear that disrupting normal splicing patterns can cause gene dysregulation and disease. For example, autosomal dominant forms with mutations in the splicing core factors can cause retinitis pigmentosa and Cerebro-Costo-Mandibular Syndrome [6,7,8,9,10], although there are few reports of mutations in core splicing elements that result in human diseases. This is likely because disruptions of basic factors in the splicing apparatus are normally more fatal compared to mutations deriving from aberrant splicing of each gene. [11]. Besides splicing core factors, many human diseases linked to defective splicing have been reported. Aberrant splicing could be caused by mutations disturbing either trans-acting regulatory genes (proteins) or cis-acting regulatory sequences, with the latter more frequently recorded.

It was recently reported that AS affects more than 85% of human protein-coding genes for cells and increases their use of genetic information [12]. Protein parts are encoded by approximately 75% alternative exons [13], and the alternate use of exons permits multiple proteins to be made from one gene, which intensifies the coding potential of the genome. It is suggested that most alternative exons encode coiled or loop regions on the protein surface, meaning AS can affect the functional interface formation of many proteins [14]. Alternative splicing generates protein isoforms with various biological properties, which can significantly affect protein–protein interaction (PPI), subcellular localization, or enzymatic ability [15]. This means that each cell might have evolved to increase the number of protein pools to respond more efficiently to harmful stress conditions brought by AS. PPIs are the main biochemical mechanisms of cellular life and are frequently disturbed in disease condition. We predict that the numbers of protein interactions (an interactome) in humans may add up to between 130,000 and 600,000 [16,17] (the number has been increasing overtime). Protein interactions in functional multi-protein complexes that have roles in basic processes such as protein transport, making an RNA copy of a gene sequence (transcription), producing proteins from mRNA (translation), interaction/communication between cells, protein modification, signaling cascades, and functional holoenzyme can be explained by the PPIs. It is not surprising that when the homeostatic condition of an organism or a single cell is interrupted (as a result of several stresses or in a disease condition), the normal patterns of PPIs can be disturbed. Furthermore, aberrant PPIs from splice variants, which are produced from mutations of a single gene in cis, could be the causes of cancerous or disease-onset cells such as those of Hutchinson-Gilford progeria syndrome (HGPS) and Small Cell Lung Cancer (SCLC) [18,19] (Figure 1 and Figure 2).

In this review, using the examples of human diseases HGPS and SCLC, we will introduce their etiology by abnormal alternative splicing and present the aberrant protein complexes that lead to the diseases. We will focus on the interfaces of protein–protein interaction to select therapeutic targets. In this article, we will highlight the proven or possible PPI interfaces that may cause disease onset and that could represent potential therapeutic targets in SCLC and HGPS.

## 2. HGPS and Present Therapeutic Agent

Hutchinson-Gilford progeria syndrome (HGPS; OMIM#176670) is an ultra-rare disease that may recapitulate some features of biological aging [20,21,22,23,24]. It has been reported that the heterozygous, de novo point mutation c.1824C>T (p.G608G) (NM_170707.3) in exon 11 of the human *LMNA* gene—which encodes Lamin A and C—mediates abnormal alternative splicing [25,26,27,28,29,30] and generates an abnormal variant protein, Progerin (50-amino-acid-deleted Lamin A), which is responsible for the nucleus deformation (Figure 1). The variant (Progerin) was also reported to be present in non-HGPS aged cells by sporadic mutation [31,32], which may be related to some aging diseases. It has been reported that Progerin accumulates in diverse human cancer cells [33,34].

Lamin A is a nuclear structural protein associated with type-V intermediate filaments; it is synthesized as a precursor named prelamin A and completed with a four-step modification. A 15-carbon farnesyl moiety is attached to the cysteine of the carboxyl-terminal (CaaX motif) by farnesyltransferase (First). Zmpste24 metallopeptidase cleaves the last three amino acids in CaaX motif (Second). A prenylprotein-specific methyltransferase carboxyl-methylates the newly exposed farnesyl cysteine (Third). Finally, the metallopeptidase gets rid of the carboxyl 15 amino acids from the farnesylated and methylated prelamin A, leading to the release of mature Lamin A without farnesyl and methyl group [35,36,37]. However, mutation-driven aberrant splicing makes Progerin keep the farnesylation at its C-terminus because of the target site deletion of the metallopeptidase. Expression of Progerin makes morphology changes in the cells, such as nucleus membrane blister, and it also induces a decrease of nucleoplasmic Lamin A [27,38]. DNA damage from intrinsic and extrinsic origin induces progressively cellular malfunction and creates vulnerabilities to developing chronic diseases related to aging. It has already been proven that reduction of extrinsic DNA damaging, such as UV protection and to stop smoking, decreased the development of age-related disease [39]. Alterations in Lamin A including HGPS have an impact both on DNA transactions and epigenetic modifications. Therefore, it is supported that Progerin-induced DNA damage or genomic instability contributes to the pathologies of aging or aging-related diseases [40]. Reduction of Progerin expression induces nuclear reformation [38,41], indicating that new or enhanced activity from Progerin is a causative factor of HGPS. Particularly, the protein levels of Progerin and Lamin A/C are remarkably decreased in induced pluripotent stem cells derived from HGPS patients [42,43]. In addition, cellular senescence markers such as nuclear deformation, tri-Methyl-Histone H3 (Lys9), and senescence-associated β galactosidase (SA–β-gal) activity, are reduced in the cells. On the other hand, differentiated HGPS cells express new senescence markers and then express Progerin [42]. Despite several papers reporting the role of this protein in cell-cycle regulation [27,44,45], premature aging [39,46,47], and senescence [41,48], how precisely Progerin causes several cellular defects and aging remains to be elucidated. However, a farnesyltransferase (FT) has been suggested as a putative therapeutic target for HGPS, and several groups showed that farnesyltransferase inhibitors (FTIs) blocked the accumulation of Progerin and reduced its amount. They potentially ameliorated the disease progression of HGPS and the processing-deficient progeroid laminopathies [49,50,51].

Lonafarnib (Zokinvy™) is an orally administrable biologically active FTI developed from Eiger BioPharmaceuticals (Palo Alto, CA, USA) with permission from Merck & Co. (Kenilworth, NJ, USA). It is produced for the medication of hepatitis D viral infections and disorders caused by mutations in genes encoding proteins of the nuclear lamina (progeroid laminopathies) [52]. Originally, Merck & Co. discovered the drug as an investigational drug for the study of cancer, but the development of lonafarnib for cancer has been canceled because of low efficacy [52]. More usefully, it was defined that it suppresses FT and prevents farnesylation. Subsequently, it reduces the aggregation of Progerin-like proteins including Progerin in the cells [51]. Two years ago, lonafarnib achieved a clinical approval in the US for the sake of reduction the risk of mortality in HGPS and for the medication of progeroid laminopathies, which are with the heterozygous Lamin A gene mutants or homozygous or compound heterozygous mutations of *ZMPSTE24* [53,54]. Indeed, a clinical trial of lonafarnib verified that it is able to increase the lifespan of HGPS patients by around two years [52], but the trial was carried out without a single-blind study. As a downside, in vitro experiments in cells showed that its treatment has cytotoxic effects, resulting in the deformation of the nuclei [55] and apoptotic cell death [56,57], which undermine its favorable effect. Currently, more effective treatments with less side effects are required for HGPS patients.

## 3. Future Therapeutic Agent for HGPS and Aging Cells

Park and his colleagues have studied age-related cancer progression and Progerin [18,34,58]. They got the idea for a possible aberrant protein complex between Lamin and Progerin that might cause the progeria disease. In proteomic studies, the Lamin A/C was identified as an interacting protein with Progerin [59,60], which made them more convinced of the possibility. Then, they speculated that the interaction between Progerin and Lamin A/C might contribute to the senescence phenotype development of HGPS and aged cells. In their report, they showed that the most important target of Progerin is Lamin A/C, not Lamin B [19]. On this basis, using the chemical library, they screened new allosteric three-chemical candidates that block the binding of Progerin and Lamin A/C through direct interaction between the chemicals and Progerin (Figure 1). The chemicals efficiently alleviated senescence properties including growth arrest, accumulation of SA–β-gal, and nucleus deformation in HGPS cells. They did not have a therapeutic effect on the Profigin-independent progeria model mice (*Zmpste24–/–*) but expanded the lifespan of the HGPS model mice (*Lmna+/G609G*), representing the in vivo relevance of blocking Progerin–Lamin A/C interaction as a new therapeutic strategy for HGPS. This study suggested that an optimization of chemicals that target the interface of Progerin-Lamin binding might help to develop a promising medication for HGPS [19]. Thereafter, Park and his colleagues tried medical chemistry using the candidate chemical and secured an advanced drug candidate, Progerinin (SLC-D011), in which the in vivo pharmacokinetics were improved. The chemical could extend the lifespan of homozygous *Lmna G609G* mice by about 10 weeks. Furthermore, heterozygous *Lmna+/G609G* mice treated by oral administration lived 14 weeks longer than the mice treated with lonafarnib (farnesyl-transferase inhibitor), which could live for only around 2 weeks longer than the control [61]. Additional comparative toxicological studies in mice, rats, and dogs demonstrated that SLC-D011 is safe to test in a human clinical trial (NCT04512963).

Recently, several articles showed that the antisense oligonucleotide or base editing that targets the point mutation c.1824C>T could rescue the aberrant splicing [62,63]. Diverse engineered Cas9 variants recognizing the sequences of modified PAM and an enhanced cleaving specificity have been evolved, and they may make it possible for us to expand the selective scope of CRISPR/Base-editors [64,65,66,67], allowing single vector delivery by an adeno-associated vector and demonstrating to be notably practical for medicinal applications. These give us an idea about the possibility of in vivo base-editing as potential therapeutics for many genetic diseases including HGPS by directly fixing their main reason. However, for human in vivo treatment, how much off-target effects on chromosomal level can be reduced in base editing steps and how efficiently viral delivery system is able to reduce side effects as like gene integration, immune activation, etc., remains to be properly addressed.

Furthermore, they extended the effect of Progerinin to Werner syndrome, a rare progressive genetic disorder that results from a functional defect of the human WRN protein, a member of the RecQ DNA helicase family [68]. The Progerin-inhibitor (SLC-D011) could ameliorate senescence in the fibroblasts and cardiomyocytes derived from Werner syndrome-iPSCs, suggesting that Progerin is able to accumulate and cause aging phenotype under WRN-deficient conditions and the phenomenon may be prevented by SLC-D011 [69]. Interestingly, under the natural aging process, Progerin can be produced and increased with age [47,70]. Natural aging and HGPS share common characteristics, such as alopecia, osteolysis, atherosclerosis, and cardiovascular complications. Cardiovascular problems are one of the most common causes of death in the elderly [71]. Progerinin (SLC-D011) could reduce fibrosis in the hearts of Lmna G609G progeroid mice and also improve their heart-beat rate [61]. These results suggest that SLC-D011 may reduce the burden of cardiovascular disease in normal aging.

## 4. SCLC, Splicing Variant DX2, and Target Molecules

Human lung cancers consist of two groups—small cell lung cancer (SCLC) and non-small cell lung cancer (NSCLC). NSCLC has three subsets: large cell lung cancer, squamous cell lung carcinoma, and adenocarcinoma. Among human lung cancers, SCLC is involved in around 20% of the cancers and has very aggressive properties. Smoking tobacco is the typical threat for SCLC, and more than 90% of patients suffering from SCLC have smoking experience [72,73,74]. SCLC has diverse genetic alterations, meaning it is genetically unstable [75,76]. Most SCLC cells have variations in chromosome 3p and frequently express mutations in oncogenes such as *RB1*, *TP53*, etc. [77,78]. The tendency toward early dissemination, rapid tumor cell growth, and neuroendocrine features are the characteristics of SCLC disease. Most patients (around 70%) show an extensive stage of disease (ES-SCLC) at the time of diagnosis, and the remaining show a limited stage of disease (LS-SCLC). SCLC has a poor prognosis, with an average survival of 10 months for ES-SCLC patients and survival of around 4 years for LS-SCLC patients [79].

Immunotherapy using checkpoint inhibitors such as Pembrolizumab, which is a humanized monoclonal antibody that binds the PD-1 receptor, and Nivolumab, which is a fully human PD-1 immune checkpoint inhibitor antibody, has shown promising medicinal effects by modulating the immune microenvironment in SCLC [80,81]. Furthermore, combinations of chemotherapy and immunotherapy and other treatment, such as SRA737 plus low dose gemcitabine with anti-PD1 antibody, have currently been in clinical trials for SCLC and other malignancies [82]. It has not been clearly understood whether the antigens allowing the immune system can discriminate cancer cells from non-cancer cells. However, recent studies have suggested that frameshifted peptides from RNA containing neo-open reading frames can be an origin of neoantigens that give rise to a cancer-specific immune response, expediting the development of novel therapeutics and selectively enhancing T cell activation against neoantigens. [83,84,85,86]. However, there is no proper anticancer drug with lower side effect against SCLC as of yet.

Aminoacyl-tRNA synthetase-interacting multifunctional protein 2 (AIMP2), also known as p38 and JTV-1, plays an important part in controlling cell fate and has an anti-proliferative role through enhancing the TGF-β-driven growth-arrest signaling [87], and it also elevates the cell death signaling via p53 or through TNF-α [88,89]. Because of this, AIMP2-deficient mice experienced neonatal mortality by lung failure resulting from the over-proliferation of pulmonary epithelial cells. Additionally, heterozygous mice, which have a reduced expression of AIMP2, were highly susceptible to tumorigenesis [90]. These results suggested that AIMP2 might be a haploinsufficient tumor suppressor with a unique working mechanism. Different from the expected roles, housekeeping or a scaffolding on the crucial multienzyme complex, it had diverse cellular functions such as acting as a p53 activator and substrate of Parkin [91,92]. The base substitution of A152 to G at its exon 2 region was found in cells transformed by carcinogens. The single mutation within splice donor or the splice acceptor sites generated exon 2 skipping and led to the accumulation of AIMP2-DX2 (also called DX2) in the cells (Figure 2). DX2, an aberrant splicing variant of AIMP2, has been shown to be at higher abundance in human lung cancer [92,93] and increased by benzopyrene [93], which exhibits genotoxic and carcinogenic effects associated with smoking [94]. From this point, Park and his colleagues speculated that splicing variant DX2 might be the critical factor of lung cancer initiation or progression, inspiring them to study the DX2 function in human lung cancer, particularly, SCLC due to the strong co-relation with smoking. They studied the expression patterns of DX2 in various human lung cancer cell lines and reported that DX2 became stable by various oncogenic forms of signaling including Her2/Neu-AKT or K-Ras activation. Then, they proved that DX2 could block the activation of oncogene-induced p14/ARF by direct binding, but not p53, supporting the report about p14/ARF inactivation without genetic mutation in SCLC [95]. Thus, they speculated that one of the plausible target points at which to treat SCLC was to prevent DX2 from binding to p14/ARF. Their next step was screening chemicals inhibiting the interaction between DX2 and p14/ARF. Through ELISA-based chemical screening, they found a small chemical inhibitor (SLCB050) blocking the interaction and additionally increasing p14/ARF and suppressing DX2 (Figure 2). It also suppressed the cell viability of SCLC cell lines with the p14/ARF-dependent manner [18]. They further showed its significantly enhanced anti-cancer effect in DX2/K-RasLA2 double-transgenic (DK) mice when treated together with GN25, inhibiting the binding of p53-Snail [96] or with Adriamycin, having antimitotic and cytotoxic activity in various cancer cells [18,97]. The chemical SLCB050 is being developed by making derivatives to improve the drug potency, efficacy, and bioavailability (BA). It is believed that the DX2 inhibitor is another example reflecting how aberrant protein–protein interfaces formed by a splicing variant in disease cells can make for an efficient drug targeting site.

## 5. Conclusions

The complex network of direct interactions between proteins, known as the interactome, and the balanced homeostatic status in cellular protein systems, known as proteostasis, have been widely recognized as important concepts for therapeutic targets for various diseases. Alternative splicing is involved in gene regulation and diversification by increasing its coding potential [12,13,14,15]. The disruption of the balance among splicing variants or neomorphic protein complexes by unusual alternative splicing variants may lead to pathological disorders. However, it is not evident whether disease induction is generated by a change in alternative splicing or whether the change just indicates a basic defect. Therefore, further studies into PPI networks, and research that defines the global proteostasis of cells, are necessary to provide a better interpretation of the fundamental cellular biochemistry and physiology of diseases. In this regard, new findings of unusual splicing variants linked to human diseases and of ways to address their pathogenic mechanisms might give us better insights into the diagnosis and therapy of diseases. Therefore, it is important to examine culprit genes where aberrant alternative splicing events are induced when exposed to stresses such as aging, genetic changes, carcinogens, etc., to support the development of pharmacological interventions against many diseases that do not have treatments to date. In this review, using the examples of SCLC and HGPS, we focused specifically on small molecules that inhibit protein–protein interactions (PPIs) by interacting directly with one protein partner from a splicing variant, rather than binding at functional sites of the other partners. We hope that this review may have meaningful ramifications to biologists developing chemical PPI inhibitors.

## Figures and Tables

**Figure 1 genes-13-00165-f001:**
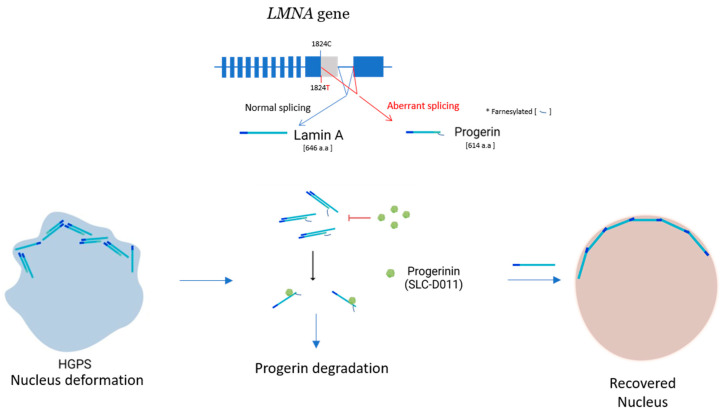
**Progerinin (SLC-D011) can ameliorate the nucleus deformation.** Progerin is produced by alternative splicing after mutation of LMNA gene. It interacts with Lamin A and generates strong heterodimers. These induce the deformation of the nucleus. Progerinin (SLC-D011) can specifically bind to Progerin and mediate it to the degradation.

**Figure 2 genes-13-00165-f002:**
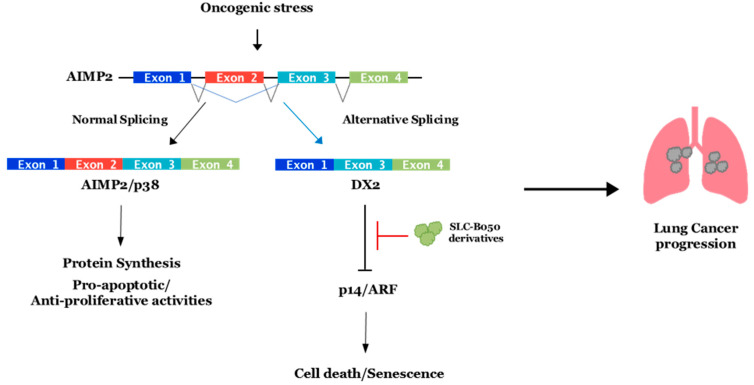
**SLCB050 inhibitor can block the interaction between DX2 and P14/ARF.** AIMP2-DX2 is produced by alternative splicing under oncogenic stress condition. It interacts with P14/ARF and inhibits cell death and senescence. Small chemical SLCB050 can specifically bind to DX2 and dissociate P14/ARF from DX2. It induces cell death or senescence dependent on P14/ARF and may contribute to blocking lung cancer progression.

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
