# Peer review of "Splicing Variants, Protein-Protein Interactions, and Drug Targeting in Hutchinson-Gilford Progeria Syndrome and Small Cell Lung Cancer"

_genes, 2022, doi:10.3390/genes13020165_

Round 1
Reviewer 1 Report
The review by Kim et al provides a well written introduction and nice overview of two examples of diseases, namely Hutchinson-Gilford Progeria Syndrome (HGPS) and Small Cell Lung Cancer (SCLC), in which mutations influence splicing driving the disease. However, there are still numerous concerns that need addressing:
- In general, this manuscript could benefit from some English grammar check, usage of Italics for Latin words such as in vitro and in vivo, as well as reduced usage of abbreviations which sometimes are even only used once in the entire manuscript.
- The reference to Figure 1 and 2 (e.g., on line 78/79) for SCLC and HGPS could be made in a more respective order.
- With respect to Figure 1, the HGPS mutation influences cleavage of LMNA, eventually leading to a shorter protein Lamin A as compared to the non-cleaved Progerin. This is not properly reflected in the Figure.
- Figure 2 could, similar to Figure 1, benefit from showing also normal splicing besides aberrant splicing.
- Line 100 “accelerated-aging disease (Fig. 1)” seems in its current form incorrect, as Figure 1 does not show any signs of accelerated aging or phenotypes of disease. This should be either added to the figure or rephrased in the text.
- Line 111 “the deleted 50 AA region in Progerin” also could be improved in the current Figure 1, as it more seems to reflect addition (red part is added in the progerin protein) rather than deletion.
- The rational of how Progerin induces premature aging is lacking in this manuscript and should be extended beyond simply listing remains to be revealed. Multiple publications exist that explain this in more detail such as Burtner C.R. and Kennedy B.K. 2010 Nat Rev Mol Cell Biol, Ghosh S. and Zhou Z 2014 Curr Opin Genet Dev, Vermeij W.P. and Hoeijmakers J.H.J. 2021 Nature, and references therein.
- The third paragraph on future therapeutic agents for HGPS now mainly focusses on self-citations to work from Park and colleagues. Rather than only focusing on chemical therapeutic options, the authors should add the recent advances of rescuing HGPS splicing aberrations by e.g., the use of base editing.
- Lastly, the paragraph on SCLC could also benefit from applying a structure like used for HGPS by adding more on potential future therapeutics using genetic tools or immune therapy with neo-open reading frame peptides.
Author Response
Reviewer 1 Comments and Suggestions for Authors
The review by Kim et al provides a well written introduction and nice overview of two examples of diseases, namely Hutchinson-Gilford Progeria Syndrome (HGPS) and Small Cell Lung Cancer (SCLC), in which mutations influence splicing driving the disease. However, there are still numerous concerns that need addressing:
Dear Reviewer 1,
We really appreciate you for reading our manuscript and giving us precious comments for better manuscript. Please look at our reply as follows;
- In general, this manuscript could benefit from some English grammar check, usage of Italics for Latin words such as in vitroand in vivo, as well as reduced usage of abbreviations which sometimes are even only used once in the entire manuscript.
- As you suggested, we corrected the points such as Latin words and reduction of abbreviation usage. And entire manuscript was checked for grammar and plagiarism using MDPI English program. Capture image of “English editing certificate” was attached on the last page.
- The reference to Figure 1 and 2 (e.g., on line 78/79) for SCLC and HGPS could be made in a more respective order.
- The reference to Figure 1 and 2 was changed to correct respective order as you suggested. It was highlighted by red color letters on page 2.
- With respect to Figure 1, the HGPS mutation influences cleavage of LMNA, eventually leading to a shorter protein Lamin A as compared to the non-cleaved Progerin. This is not properly reflected in the Figure.
- In Figure 1, we added the amino acid sizes of mature LaminA and Progerin to clarify the length difference (Aberrant isoform of LMNA).
- Figure 2 could, similar to Figure 1, benefit from showing also normal splicing besides aberrant splicing.
- Normal spliced AIMP2 was added to Figure 2 as you suggested.
- Line 100 “accelerated-aging disease (Fig. 1)” seems in its current form incorrect, as Figure 1 does not show any signs of accelerated aging or phenotypes of disease. This should be either added to the figure or rephrased in the text.
- We rephrased “accelerated-aging disease” to “nucleus deformation” for better explanation of Figure 1 as you suggested. It was highlighted by red color letters on page 4.
- Line 111 “the deleted 50 AA region in Progerin” also could be improved in the current Figure 1, as it more seems to reflect addition (red part is added in the progerin protein) rather than deletion.
- For a better explanation about Progerin protein, we deleted the red color and added the tag for Farnesylation to its C-terminus. Please check the updated Figure 1.
- The rational of how Progerin induces premature aging is lacking in this manuscript and should be extended beyond simply listing remains to be revealed. Multiple publications exist that explain this in more detail such as Burtner C.R. and Kennedy B.K. 2010 Nat Rev Mol Cell Biol, Ghosh S. and Zhou Z 2014 Curr Opin Genet Dev, Vermeij W.P. and Hoeijmakers J.H.J. 2021 Nature, and references therein.
- We added the rationale in the page 4 as follows and included new corresponding references in manuscript as you suggested;
“DNA damaging from intrinsic and extrinsic origin induce progressively cellular malfunction, and made vulnerable to developing the chronic diseases related to aging. It has been already proven that reduction of extrinsic DNA damaging, such as UV protection and stop smoking, decreased development of age-related disease [39]. Alterations in Lamin A including HGPS have impact both on DNA transactions and epigenetic modifications. Therefore, it is supporting that Progerin-induced DNA damage or genomic instability contributes to the pathologies of aging or aging-related diseases [40].”
“premature aging [39, 46, 47]”
They were highlighted by red color letters on page 4.
- The third paragraph on future therapeutic agents for HGPS now mainly focusses on self-citations to work from Park and colleagues. Rather than only focusing on chemical therapeutic options, the authors should add the recent advances of rescuing HGPS splicing aberrations by e.g., the use of base editing.
- As you suggested, we added the recent advances about base editing and included new corresponding references as follows;
“Recently, several articles showed the antisense oligonucleotide or base editing that targets the point mutation c.1824C>T could rescue the aberrant splicing [62, 63]. A diverse engineered Cas9 variants recognizing the sequences of modified PAM and an enhanced cleaving specificity have been evolved, and they may make it possible for us to expand the selective scope of CRISPR/Base-editors [64-67], making us allowing single vector delivery by adeno-associated vector and demonstrating notably practical for medicinal applications. These give an idea about the possibility of in vivo base-editing as a potential therapeutics for many genetic diseases including HGPS by directly fixing their main reason. However, for human in vivo treatment, how much off-target effects on chromosomal level can be reduced in base editing steps and how efficiently viral delivery system is able to reduce side effects as like gene integration, immune activation, etc. remain to be still properly addressed.”
It was highlighted by red color letters on page 5.
- Lastly, the paragraph on SCLC could also benefit from applying a structure like used for HGPS by adding more on potential future therapeutics using genetic tools or immune therapy with neo-open reading frame peptides.
- As you suggested, we added two more therapeutics and included new corresponding reference as follows;
“Immunotherapy using checkpoint inhibitors such as Pembrolizumab, which is a humanized monoclonal antibody that binds the PD-1 receptor, and Nivolumab, which is a fully human PD-1 immune checkpoint inhibitor antibody, has shown promising medicinal effects by modulating the immune microenvironment in SCLC [81, 82]. Furthermore, combinations of chemotherapy and immunotherapy and other treatment, such as SRA737 plus low dose gemcitabine with anti-PD1 antibody, have been currently in clinical trials for SCLC and other malignancies [83]. It has been not clearly understood whether the antigens allowing the immune system can discriminate cancer cells from non-cancer cells. However, recent studies have suggested that frameshifted peptides from RNA containing neo open reading frames can be an origin of neoantigens that give rise to cancer-specific immune response, expediting the development of novel therapeutics selectively enhancing T cell activation against neoantigens. [84-87]. However, there is no proper anticancer drug with lower side effect against SCLC as of yet.”
It was highlighted by red color letters on page 6.

Reviewer 2 Report
The manuscript "Splicing variants, protein-protein interactions, and drug targeting in Hutchinson-Gilford progeria syndrome and Small cell lung cancer" by Kim and co-authors discusses the possible impact of aberrant splicing on the development of pathological conditions exemplified by Hutchinson-Gilford progeria syndrome and Small cell lung cancer. The authors believe that in the cases described the main mechanism behind these diseases is an aberrant protein-protein interaction and suggest a generalized approach for therapeutic intervention based on the use of small molecules targeting the interface between mutated proteins and their partners. I wouldn't call the manuscript a review as it in fact deals with only two cases of many more existing situations. Moreover, I do not see a specific link between alternative splicing and aberrant protein-protein interactions, as point mutations can cause similar effects. To draw more sound conclusions I would recommend extending the scope of the manuscript by including a more systematic analysis of the data obtained by others. On top of that, serious language editing and proofreading are required, as some passages are hard to understand.
Author Response
Reviewer 2 Comments and Suggestions for Authors
The manuscript "Splicing variants, protein-protein interactions, and drug targeting in Hutchinson-Gilford progeria syndrome and Small cell lung cancer" by Kim and co-authors discusses the possible impact of aberrant splicing on the development of pathological conditions exemplified by Hutchinson-Gilford progeria syndrome and Small cell lung cancer. The authors believe that in the cases described the main mechanism behind these diseases is an aberrant protein-protein interaction and suggest a generalized approach for therapeutic intervention based on the use of small molecules targeting the interface between mutated proteins and their partners. I wouldn't call the manuscript a review as it in fact deals with only two cases of many more existing situations. Moreover, I do not see a specific link between alternative splicing and aberrant protein-protein interactions, as point mutations can cause similar effects. To draw more sound conclusions I would recommend extending the scope of the manuscript by including a more systematic analysis of the data obtained by others. On top of that, serious language editing and proofreading are required, as some passages are hard to understand.
Dear Reviewer 2,
We appreciate you for reading our manuscript and giving us precious comments for better manuscript. Using MDPI English edit program, the entire manuscript was examined for grammar and plagiarism, and edited for proper English. Capture image of “English editing certificate” was attached on the last page. And then, we try to focus on extending the scope of the manuscript by including a more systematic analysis of the data by others.
First, because the rational of how Progerin induces premature aging is lacking in the manuscript, we added the rationale in the page 4 with new corresponding references as follows;
“DNA damaging from intrinsic and extrinsic origin induce progressively cellular malfunction, and made vulnerable to developing the chronic diseases related to ageing. It has been already proven that reduction of extrinsic DNA damaging, such as UV protection and stop smoking, decreased development of age-related disease [39]. Alterations in Lamin A including HGPS have impact both on DNA transactions and epigenetic modifications. Therefore, it is supporting that Progerin-induced DNA damage or genomic instability contributes to the pathologies of aging or aging-related diseases [40].”
“premature aging [39, 46, 47]”
They were highlighted by red color letters on page 4.
Second, because we only focused on chemical therapeutic options, we added the recent advances of rescuing HGPS splicing aberrations by the use of genetic tools and included new corresponding references as follows;
“Recently, several articles showed the antisense oligonucleotide or base editing that targets the point mutation c.1824C>T could rescue the aberrant splicing [62, 63]. A diverse engineered Cas9 variants recognizing the sequences of modified PAM and an enhanced cleaving specificity have been evolved, and they may make it possible for us to expand the selective scope of CRISPR/Base-editors [64-67], making us allowing single vector delivery by adeno-associated vector and demonstrating notably practical for medicinal applications. These give an idea about the possibility of in vivo base-editing as a potential therapeutics for many genetic diseases including HGPS by directly fixing their main reason. However, for human in vivo treatment, how much off-target effects on chromosomal level can be reduced in base editing steps and how efficiently viral delivery system is able to reduce side effects as like gene integration, immune activation, etc. remain to be still properly addressed.”
It was highlighted by red color letters on page 5.
Third, as a structure used for HGPS by adding more on potential future therapeutics were added in manuscript, we added recent promising therapies using immune therapy on SCLC and possible intervention as like neo-open reading frame peptides in cancers.
- “Immunotherapy using checkpoint inhibitors such as Pembrolizumab, which is a humanized monoclonal antibody that binds the PD-1 receptor, and Nivolumab, which is a fully human PD-1 immune checkpoint inhibitor antibody, has shown promising medicinal effects by modulating the immune microenvironment in SCLC [81, 82]. Furthermore, combinations of chemotherapy and immunotherapy and other treatment, such as SRA737 plus low dose gemcitabine with anti-PD1 antibody, have been currently in clinical trials for SCLC and other malignancies [83]. It has been not clearly understood whether the antigens allowing the immune system can discriminate cancer cells from non-cancer cells. However, recent studies have suggested that frameshifted peptides from RNA containing neo open reading frames can be an origin of neoantigens that give rise to cancer-specific immune response, expediting the development of novel therapeutics selectively enhancing T cell activation against neoantigens [84-87]. However, there is no proper anticancer drug with lower side effect against SCLC as of yet.”
It was highlighted by red color letters on page 6.
We hope the edited and updated version make you more convinced for our manuscript. Please look at the modified Figures and texts and please give any additional comments to improve the manuscript. We are ready for replying to precious questions. And we hope to express our gratitude for spending your time on our manuscript.

Round 2
Reviewer 1 Report
The authors have substantially improved the manuscript and therefore I would endorse this manuscript for publication.